# Sex- and Genotype-Dependent Nicotine-Induced Behaviors in Adolescent Rats with a Human Polymorphism (rs2304297) in the 3′-UTR of the *CHRNA*6 Gene

**DOI:** 10.3390/ijms23063145

**Published:** 2022-03-15

**Authors:** Anjelica Cardenas, Yu Bai, Yasamin Hajy Heydary, Jiaqi Li, Frances M. Leslie, Shahrdad Lotfipour

**Affiliations:** 1Department of Pharmaceutical Sciences, College of Health Sciences, University of California, Irvine, CA 92697, USA; yheydary@uci.edu (Y.H.H.); fmleslie@uci.edu (F.M.L.); shahrdad@uci.edu (S.L.); 2Department of Biological Sciences, University of California, Irvine, CA 92697, USA; yu.bai@emory.edu (Y.B.); jiaql31@uci.edu (J.L.); 3Department of Emergency Medicine, School of Medicine, University of California, Irvine, CA 92697, USA; 4Pathology and Laboratory Medicine, School of Medicine, University of California, Irvine, CA 92697, USA

**Keywords:** *CHRNA6*, locomotor activity, anxiety-like behavior, adolescent, pharmacogenetics

## Abstract

In human adolescents, a single nucleotide polymorphism (SNP), rs2304297, in the 3′-UTR of the nicotinic receptor subunit gene, *CHRNA*6, has been associated with increased smoking. To study the effects of the human *CHRNA6* 3′-UTR SNP, our lab generated knock-in rodent lines with either C or G SNP alleles. The objective of this study was to determine if the *CHRNA6* 3′-UTR SNP is functional in the knock-in rat lines. We hypothesized that the human *CHRNA6* 3′-UTR SNP knock-in does not impact baseline but enhances nicotine-induced behaviors. For baseline behaviors, rats underwent food self-administration at escalating schedules of reinforcement followed by a locomotor assay and a series of anxiety tests (postnatal day (PN) 25-39). In separate cohorts, adolescent rats underwent 1- or 4-day nicotine pretreatment (2×, 30 μg/kg/0.1 mL, i.v.). After the last nicotine injection (PN 31), animals were assessed behaviorally in an open-field chamber, and brain tissue was collected. We show the human *CHRNA6* 3′-UTR SNP knock-in does not affect food reinforcement, locomotor activity, or anxiety. Further, 4-day, but not 1-day, nicotine exposure enhances locomotion and anxiolytic behavior in a genotype- and sex-specific manner. These findings demonstrate that the human *CHRNA6* 3′-UTR SNP is functional in our in vivo model.

## 1. Introduction

Initiation and establishment of nicotine-containing product use typically occurs during adolescence [1]. In recent years, an escalation of electronic (e)-cigarette use has been observed among middle- and high-school students in the United States [2]. Adolescent nicotine exposure can interfere with the maturation process of the brain resulting in long-term behavioral and neurochemical consequences [3]. With the rise of teen e-cigarette use, there is also an increase in the frequency in which e-cigarettes are used, suggesting adolescents are developing a strong dependence to nicotine [4]. Therefore, it is critical to identify at-risk populations and potential therapeutic targets to prevent and treat nicotine dependence and other effects.

Heritable risk factors are associated with adolescent nicotine use [1]. Single nucleotide polymorphisms (SNPs), in genes coding for nicotinic acetylcholine receptors (nAChRs) subunits, have been implicated in adolescent nicotine dependence [5]. Neuronal nAChRs are critical during the developmental period of adolescence [6] and are expressed in brain regions important for reward [7,8]. Nicotine-induced activation of nicotinic receptors regulates dopamine (DA) release in adolescents [9] and is important in reward processing and drug reinforcement [10,11]. Genetic association studies have identified a single nucleotide polymorphism (SNP), rs2304297, in the 3′-UTR of the alpha(α)6 nAChR subunit gene (*CHRNA6*) at base pair 123. The α6 nAChR subunit SNP is associated with increased cigarette smoking and drug experimentation during adolescence [12,13,14,15,16,17]. Mechanisms underlying how the *CHRNA6* 3′-UTR SNP mediates adolescent substance use are not known. However, we know that 3′-UTRs are important for mRNA processes including mRNA stability, expression, and localization [18,19].

α6 nAChR subunits are selectively expressed in α6β2β3 and α6α4β2β3 pentameric conformations, restricted primarily to the DA neurons in the mesostriatal pathway, and reach peak mRNA expression in the ventral tegmental area (VTA) and substantia nigra (SN) during adolescence [9,20,21]. The VTA regulates drug reward, locomotor activity, and anxiety-like behaviors [22,23]. The role of α6-containing (α6*) nAChRs in nicotine-induced behaviors has been assessed previously using pharmacological approaches and genetic animal models where α6* nAChRs in the VTA have been found to be necessary for nicotine-induced locomotion [23,24,25,26,27] and DA release [20,24,28] as well as nicotine self-administration and reward [28,29,30,31]. For nicotine-induced locomotion, these effects are specific to α6β2-containing nAChRs in the VTA [26]. However, α6* nAChR studies were completed in adult, male mice and rats. 

To study the human *CHRNA6* 3′-UTR SNP in vivo, our lab generated a novel, humanized rodent line. By replacing the *CHRNA6* 3′-UTR of the rat with the human *CHRNA6* 3′-UTR, our mutant rodents represent a translational model of α6 nAChRs. The objective of this study was to determine whether the human *CHRNA6* 3′-UTR knock-in is functional using the human *CHRNA6* 3′-UTR rat lines. Gene-drug association studies (i.e., pharmacogenetics) are necessary to understand how genetics affect the response to drugs [32,33,34]. Given that genetic factors influence tobacco product use [35], this study assessed how a SNP in the 3′-UTR of the *CHRNA*6 gene affects nicotine-induced behaviors during a critical time in development, adolescence.

Clinical studies identify the SNP allele, G, as the major allele associated with adolescent tobacco use, progression, and dependence [12,13,14,15,16,17]. Therefore, we hypothesized that the *CHRNA6* 3′-UTR SNP knock-in would not impact baseline behaviors but G-carrier adolescent rodents would have increased nicotine-induced locomotion and anxiety-like behavior and α6 mRNA expression as compared to C-carrier rodents. To test our hypothesis, humanized *CHRNA6* 3′-UTR SNP rats underwent a battery of behavioral tests that are not known to be impacted by α6* nAChRs, including food reward, locomotor activity, and anxiety-like behavior [24,25,30,31]. To determine the role of the *CHRNA6* 3′-UTR SNP in nicotine-induced locomotion and anxiety-like behavior, we used an acute (1 × exposure) and a well-established sub-chronic (4-day) nicotine pretreatment paradigm to model nicotine initiation in adolescence [36,37] to evaluate nicotine-induced enhancement of locomotion and anxiety-like behaviors in humanized *CHRNA6* 3′-UTR SNP rodents. Lastly, we assessed whether nicotine exposure alters α6 mRNA expression in our 4-day treated adolescents via in situ hybridization in the humanized *CHRNA6* 3′-UTR SNP rats. 

## 2. Results

### 2.1. Generation and Validation of Humanized CHRNA6 3′-UTR SNP Rodents

Humanized *CHRNA6* 3′-UTR SNP (rs2304297) rodents were generated via CRISPR/Cas9 gene targeting/editing methods. The 3′-UTR of the rat *CHRNA6* gene was replaced with the human *CHRNA6* 3′-UTR containing SNP, rs2304297 (Figure 1). SNP rs2304297 is a C to G polymorphism located at nucleotide position 123 of the *CHRNA6* 3′-UTR. Cas9 mRNA, CRISPR single guide(sg)RNA, and donor vectors were co-injected into fertilized Sprague-Dawley rat eggs and implanted into surrogate dams. Pups were genotyped via PCR followed by DNA sequence analysis. Identified founders (F0) were bred, and their positive offspring (F1) were subsequently crossbred to generate heterozygous α6^C/G^ rodents (F2). Our heterozygous rodents were then bred to generate homozygous major α6^GG^ and homozygous minor α6^CC^ human *CHRNA6* 3′-UTR SNP rodents.

### 2.2. The Human CHRNA6 3′-UTR Knock-In Does Not Impact Baseline Behavior

To determine whether the humanized *CHRNA6* 3′-UTR SNP knock-in impacts baseline behaviors (e.g., food reward, locomotor activity, anxiety-like behavior, Figure 2A), we first evaluated mean responses during food self-administration at Fixed Ratio (FR)1 and then escalated to higher schedules of reinforcement at FR2, FR5, and Progressive Ratio (PR) (Figure 2B,C). Overall ANOVA for FR1 illustrates a main effect for response (F_1,28_ = 32.12, *p* < 0.0001) and day (F_4,112_ = 13.68, *p* < 0.0001) with a response × day interaction (F_4,112_ = 22.53, *p* < 0.0001). For FR2 and FR5, we observe a main effect for response (F_1,28_ = 40.76, *p* = 0.0001 and F_1,28_ = 56.57, *p* = 0.0001, respectively). Although we did not identify interactions for every measure, we separated the data by genotype, response, and day to demonstrate no SNP genotype effects. Post hoc analysis shows that both α6^CC^ and α6^GG^ rodents have a preference for reinforced over non-reinforced responding after day 3 of FR1, and this effect persists at higher schedules of reinforcement (*p* < 0.05, Figure 2B). Overall ANOVA for mean break point during food reinforcement at PR illustrated a within effect for day (F_1,28_ = 11.19, *p* = 0.0024) and sex × day (F_1,28_ = 6.11, *p* = 0.02) interactions. Post hoc analysis shows that sex effects did not persist by the second day of PR (Figure 2C). Taken together, our data suggest the human *CHRNA6* 3′-UTR knock-in does not affect natural food reward, even at higher schedules of reinforcement.

Following food self-administration, we determined whether the human *CHRNA6* 3′-UTR SNP knock-in impacts baseline locomotion and anxiety-like behaviors using an open field test, elevated plus maze (EPM), and light/dark box test (LDT). For locomotor activity and the open field test, no main or interactive effects were observed for total ambulatory counts (Figure 2D) or % center time (Figure 2E). When assessing anxiety-like behavior via EPM (% open arm time), no main or interactive effects for sex or genotype were identified (Figure 2F). For LDT (% light side time), our results show that females were less anxious than males (F_1,28_ = 6.44, *p* = 0.017), independent of genotype (Figure 2G). No other main or interactive effects were observed.

### 2.3. Nicotine Increases Locomotor Activity in Humanized CHRNA6 3′-UTR SNP Rats

To test the function of the human *CHRNA6* 3′-UTR SNP knock-in in vivo, we assessed acute (1-day) and sub-chronic (4-day) nicotine-induced locomotor activity and anxiety-like behavior. For locomotor activity, overall ANOVA for acute nicotine exposure illustrated enhanced locomotion (pretreatment effect, F_1,58_ = 4.50, *p* = 0.0382). No other main or interactive effects were observed. Data were collapsed by sex and separated by genotype and pretreatment (Figure 3A). For sub-chronic nicotine exposure, overall ANOVA shows nicotine-induced enhancement of locomotion (F_1,60_ = 20.31, *p* = 0.0001) and a genotype × sex (F_1,60_ = 6.24, *p* = 0.015) interaction. Since we identified interactions for every measure, we separated the data by genotype, sex, and pretreatment (Figure 3B). For females, we observed nicotine-enhanced locomotion in α6^CC^ rats (*p* = 0.0002) as compared to nicotine-treated α6^GG^ rats (*p* = 0.018) and saline-treated α6^CC^ rats (Figure 3B). In males, nicotine increased locomotion in α6^GG^ rats as compared to saline-treated α6^GG^ rats (*p* = 0.0488, Figure 3B). Taken together, our results show that sub-chronic, but not acute, nicotine exposure leads to sex- and genotype-dependent enhancement of locomotion in humanized *CHRNA6* 3′-UTR SNP rodents.

### 2.4. Nicotine Increases Anxiolytic Behavior in Humanized CHRNA6 3′-UTR SNP Rats

For anxiety-like behavior, we assessed the percent (%) time spent in the center of the open-field chamber (Figure 4). Acute nicotine exposure enhanced overall anxiolytic behavior (pretreatment effect, F_1,57_ = 12.80, *p* = 0.0007). Additionally, a genotype × pretreatment (F_1,57_ = 7.90, *p* = 0.0068) interactive effect was shown. Data were collapsed by sex and separated by genotype and pretreatment. Post hoc analysis showed that acute nicotine-exposed α6^CC^ rats have enhanced anxiolytic behavior versus saline-treated α6^CC^ and nicotine-treated α6^GG^ rats (*p* = 0.0002 and *p* = 0.002, respectively, Figure 4A). For sub-chronic nicotine exposure, there was a main effect of sex (F_1,60_ = 14.22, *p* = 0.0004) and pretreatment (F_1,60_ = 11.43, *p* = 0.0013) as well as sex × genotype (F_1,60_ = 9.99, *p* = 0.0025) and sex × genotype × pretreatment (F_1,60_ = 5.29, *p* = 0.025) interactions. Data were separated by every measure. Post hoc analysis showed nicotine-induced anxiolytic behavior in female α6^CC^ (*p* = 0.0026) and male α6^GG^ (*p* = 0.019) rats (Figure 4B). Furthermore, nicotine-treated female α6^CC^ and male α6^GG^ rats illustrated enhanced anxiolytic behavior as compared to nicotine-treated female α6^GG^ and male α6^CC^ rats (*p* = 0.036 and *p* = 0.0048, respectively, Figure 4B). Overall, our findings show that sub-chronic, but not acute, nicotine exposure leads to sex- and genotype-dependent enhancement of anxiolytic behavior in humanized *CHRNA6* 3′-UTR SNP rodents.

### 2.5. Nicotine Pretreatment Does Not Alter mRNA Expression in Humanized CHRNA6 3′-UTR SNP Rats

To determine whether a nicotine-induced increase in α6 expression may underlie our observed behaviors, we quantified α6 mRNA expression in the behaviorally matched tissue of our adolescent knock-in rodents. Since α6 nAChR subunits are expressed in the DA neurons of the ventral midbrain we assessed α6 expression in the VTA, SN, and interpeduncular nucleus (IPN) [9,20,21]. The overall ANOVA of disintegrations per minute (dpm)/mg in the mesostriatal regions revealed a main effect for the region (F_3,75_ = 28.51, *p* = 0.0001). Although no other main or interactive effects were identified, we separated data by region, pretreatment, and genotype for clarity (Figure 5). No additional effects were observed in post hoc analyses. Together, our data suggest that adolescent nicotine exposure does not interact with sex or SNP genotype to influence α6 mRNA expression.

## 3. Discussion

In this present study, we add to the literature by examining the role of an α6 nAChR subunit polymorphism during adolescence in nicotine-induced behaviors using a novel rodent model. We successfully showed that the human *CHRNA6* 3′-UTR SNP is functional in vivo. Data collected support the hypothesis that our genetic knock-in of the human *CHRNA6* 3′-UTR does not impact baseline behaviors. When assessing acute versus sub-chronic nicotine-induced behaviors, we showed genotype- and sex-dependent effects. However, we did not find nicotine-induced changes in mRNA expression. Taken together, differences in nicotine-induced locomotion in α6^GG^ compared to α6^CC^ animals suggest that the human *CHRNA6* 3′-UTR SNP rodent line is a valid, translational animal model.

### 3.1. Baseline Behaviors

Our baseline data are in accord with studies that show α6* nAChRs do not alter anxiety [30], food reward [31], and general locomotion [24,25] in rodents. Independent of sex, our humanized rats acquire and maintain food self-administration, even at higher schedules of reinforcement. Using a battery of anxiety-like tests [38], we assessed different aspects of anxiety-like behavior [38]. The open field test and EPM measure exploratory behavior, whereas the LDT measures aversion to an open, well-lit area [38,39]. We showed that males display greater baseline anxiety-like behavior than females, independent of SNP genotype for the LDT but not for EPM or open field test measures. These data are in contrast to a study that found no sex effects for the LDT in adolescent control rats [40]. Our observed sex effect for LDT may result from differences in methodology, such as the age at time of test, the battery of the test prior to LDT, and the order of anxiety-like behavior tests in this study. Of note, few studies have assessed sex and age when modeling anxiety, which highlights the need for more studies that include all groups when testing anxiety-like behaviors [39,41].

### 3.2. Acute Nicotine Effects

To assess nicotine-induced behavioral effects in humanized *CHRNA6* 3′-UTR SNP rodents we used acute (1-day) and sub-chronic (4-day) nicotine exposure paradigms [36,37], which model adolescent nicotine initiation and/or experimentation in humans. Our findings are consistent with adolescent wild-type rat studies, which show low-dose nicotine exposure during early adolescence is sufficient to enhance nicotine-induced locomotion and anxiolytic behavior after nicotine infusion [37,42]. Our data are in line with other α6* animal model studies that have shown acute nicotine exposure is sufficient for α6* nicotinic receptor-mediated locomotor enhancement [23,24,25,26,27]. In the open field test, acute nicotine exposure further increased anxiolytic behavior in the α6^CC^ and nicotine-treated α6^GG^ animals. Unexpectedly, the enhanced nicotine effects were not in the SNP genotype we hypothesized.

### 3.3. Sub-Chronic Nicotine Effects

In sub-chronically treated humanized *CHRNA6* 3′-UTR SNP rats, significant nicotine effects on locomotion and anxiety-like behavior were sex- and genotype-dependent. Nicotine-induced locomotion has been shown previously in adolescent WT rats using the 4-day nicotine pretreatment paradigm, however, only in male rats [37]. α6^CC^ females and α6^GG^ males displayed nicotine-induced enhanced locomotor and anxiolytic behavior compared to their saline-treated counterparts. Furthermore, nicotine-induced locomotion and anxiolytic behavior in α6^CC^ females was significantly higher than α6^GG^ females. In contrast, nicotine-induced anxiolytic behavior in males was significantly higher in α6^GG^ as compared to α6^CC^ animals. Nicotine-induced anxiety-like behavior has not been extensively evaluated in α6* nAChR receptor animal models.

Neuronal nAChR α6 subunit mRNA expression has been found to be restricted to the nuclei of catecholaminergic neurons in rodents [43]. Moreover, mRNA expression reaches peak expression during adolescence in the VTA and SN, brain regions implicated in the modulation of nicotine-induced locomotion and reward [9,43]. We did not identify genotype-, sex-, or nicotine-dependent alterations in mRNA expression. Regardless, genetic mRNA expression levels do not necessarily correspond to receptor protein expression [44,45]. To assess protein expression in the humanized rodents, future studies will need to forego commercial antibodies because currently available α6 nAChR subunit antibodies lack specificity [46,47]. Alternatively, a radioligand binding assay in the brain tissue of human *CHRNA6* 3′-UTR SNP rodents should be pursued to quantify α6 receptor (protein) expression.

### 3.4. Sex Effects

Bidirectional sex-dependent effects have been reported previously for transcriptional regulation of α6* nAChRs via protein kinase C epsilon in adult mice [48,49]. Although sexual differentiation takes place in the brain during adolescence, our early adolescent nicotine exposure effects are likely independent of puberty influences [3,50,51]. However, in addition to nicotine, sex steroids and steroid metabolites interact with nAChRs to induce age-, sex-, and dose-dependent effects [6,52], which may be further influenced by genetics. Increased nicotine-stimulated DA release, striatal DA receptors, and pruning may furthermore be affected by genotype in addition to innate sex differences in the DA system [9,53,54]. Adolescent clinical studies did not observe or report sex effects for the human *CHRNA6* 3′-UTR SNP [12,14,17], but one study did find that sex differences increased with age [15]. Additional studies are needed to better understand the mechanisms underlying nicotine-induced sex- and SNP genotype-dependent effects in the humanized *CHRNA6* 3′-UTR SNP rats.

### 3.5. Mechanisms

Nicotine has previously been shown to interact with genetics to influence behavior [35]. Similarly, SNPs have been shown to affect response to therapy and behavior [32,55]. However, the mechanisms underlying the human *CHRNA6* 3′-UTR SNP nicotine-induced behavioral phenotypes are unknown, and methods for studying 3′-UTR functions are lacking [19]. It is known that RNA-binding proteins (RBP) and micro(mi)RNAs interact with 3′-UTRs to influence mRNA stability, localization, and translation [19]. In one study, a SNP in the 3′-UTR of the prodynorphin gene was shown to alter miRNA binding and influence positive-reinforcement behavior [55]. Future studies should assess human miRNA effects on the *CHRNA6* 3′-UTR SNP in vitro. Many miRNAs that bind to or near the site of our SNP of interest are specific to humans and not present in the rat. As the field advances, RBP binding motifs can be identified and tested in our genetic rodent model as they become known. In the meantime, preclinical studies can be pursued to better understand the role of the *CHRNA6* 3′-UTR SNP in adolescent nicotine-induced behaviors. While nicotine-induced locomotion is closely associated with self-administration of addictive substances, future studies should include nicotine self-administration, extinction, and reinstatement in humanized *CHRNA6* 3′-UTR SNP rats. In addition, baseline and nicotine-induced DA release and nicotine-induced subsequent drug reinforcement (i.e., gateway hypothesis) and acquisition of other drugs of abuse should be assessed in the humanized *CHRNA6* 3′-UTR SNP rats. Using the 4-day nicotine exposure paradigm, studies have shown nicotine enhances subsequent cocaine [36,56,57,58], methamphetamine [56,59], ethanol [56], and fentanyl intake [58,60,61]. Furthermore, nicotine potency and efficacy may be shifted in the humanized *CHRNA6* 3′-UTR SNP rodents, and future studies should implement a dose–response curve.

## 4. Conclusions

With the high incidence of nicotine initiation and use during adolescence, it is vital to identify vulnerable populations. In this study, we generated and validated a novel, humanized rodent model to study the effects a human SNP in the 3′-UTR of the *CHRNA6* gene in nicotine-induced locomotor activity and anxiety-like behavior. Whereas no group differences in baseline behaviors were observed, we identified variability in response to the drug (i.e., nicotine) due to the SNP genotype with sex as an additional predictor of response [32]. We further showed that observed behavioral alterations were not mediated by altered α6 nAChR subunit mRNA expression. Overall, our results highlight the nuances and importance of including age, biological sex, environmental influences, and genetics to evaluate substance use. By including all groups, potential targeted therapeutic strategies may be developed for identified or individual at-high-risk populations.

## 5. Materials and Methods

### 5.1. Generation of Human CHRNA6 3′-UTR SNP Rodents

Human *CHRNA6* 3′-UTR SNP knock-in rodents were designed and created with CRISPR/Cas9 gene targeting/editing techniques by Cyagen Biosciences (Guanzhou, China). Donor vectors and sgRNA were designed to target the 3′-UTR (95 nucleotides) of the rat *CHRNA*6 gene (GenBank accession number: NM_057184.1; Ensembl: ENSRNOG00000012283). Double stranded DNA breaks were repaired via homologous recombination. Donor vectors contained the human *CHRNA6* 3′-UTR (559 nucleotides) with either the minor SNP rs2304297 allele, C, or major SNP rs2304297 allele, G, at nucleotide position 123 (Figure 1) and replaced the rodent 3′-UTR. In addition to the human *CHRNA6* 3′-UTR, the vector contained left (766 bp) and right (913 bp) homology arms. Human *CHRNA6* 3′-UTR SNP knock-in pups were genotyped by PCR, followed by DNA sequence analysis. DNA sequence analysis confirms the C or G SNP allele at position 123 of the *CHRNA6* 3′-UTR, and good sequencing for the homologous identity of the upstream and downstream 3′-UTR knock-in sites and Cas9 cleavage in top five off-target sites were confirmed. The offspring of mated human *CHRNA6* 3′-UTR SNP rats were genotyped by Transnetyx (Cordova, TN, USA).

### 5.2. Animals

Male and female wild-type (WT) Sprague–Dawley rats were purchased from Charles River and bred in house with human *CHRNA6* 3′-UTR SNP rodents. Juveniles were weaned at postnatal day (PN) 21. All animals were handled for 3 days prior to experimentation and group-housed in a controlled 12-h light–dark cycle (lights on 0700–1900) in an AAALAC-accredited vivarium. Food and water were provided *ad libitum* except when indicated. Animals were weighed daily to ensure the maintenance of normal growth. To avoid potential litter effects, one pup per litter per experimental group was used for data collection. Animals were allocated to experimental groups using a random sequence generator. All experiments were carried out in accordance with the Institutional Animal Care and Use Committee at the University of California, Irvine, CA, USA.

### 5.3. Food Reinforcement

Adolescent rats (PN25–PN36) were trained to lever press for food pellets on a Fixed Ratio (FR)1TO20 schedule for 6 days followed by FR 2, FR 5, and progressive ratio (PR) schedules for 2 days each. Before each round, the rats were food deprived and given 15 g of food/animal after each round. During the session, reinforced and non-reinforced lever pressing was recorded.

### 5.4. Locomotor Apparatus

Locomotion activity was tested by using four identical open-field activity chambers measuring 43.2 × 43.2 × 30.5 cm, which connected to a common interface and computer (Med Associates, St. Albans, VT, USA). There are two adjacent sides of sixteen evenly located infrared monitors recording the horizontal movement. To ensure fair comparisons between ages, the number of horizontal beam fractures recorded was normalized to the length of the rat. Ambulatory counts and time spent in the center of the open field were recorded automatically and analyzed after the conclusion of each experiment.

### 5.5. Baseline Locomotor Activity & Open Field Test

On PN37, rats were placed in a novel chamber. The locomotor activity was recorded at 5 min intervals during the subsequent 30-min test period.

### 5.6. Elevated Plus Maze

On PN 38, animals were subjected to an anxiety-like test in the Elevated Plus Maze (EPM). The EPM is a plus-shaped acrylic maze with two open (50 cm in length and 10 cm in width) and two closed arms (50 cm in length, 10 cm in width, and 31 cm in height) extending out from an open central junction (10 × 10 cm). The apparatus was 50 cm above the floor. Following a 30 min habituation, animals were placed in the center of the maze facing a closed arm and allowed to explore the apparatus for five min. The time spent in each arm or the junction, as well as total closed or open arm entries, was recorded (Med-PC IV, Med Associates Inc., St. Albans, VT, USA).

### 5.7. Light/Dark Box Test

On PN 39, animals underwent testing of anxiety-like behavior in a light/dark box according to previously published methods. The light/dark box consists of an open-field activity chamber (43.38 × 43.38 × 30.28 cm) containing a dark box insert (44.4 × 22.9 × 30.5 cm) with an open door to allow for movement between the enclosed dark side and the light side. The activity chamber was connected to a common interface and computer (Med Associates Inc., St. Albans, VT, USA). Following a 30 min habituation period, animals were placed in the dark side of the apparatus and the lid was closed. Time spent in the light and dark sides of the chamber, as well as distance travelled, was recorded for five min.

### 5.8. Drugs & Reagents

Nicotine tartrate (Glentham Life Sciences, Corsham, Wiltshire, UK) was calculated as a base, dissolved in saline, and pH adjusted to 7.2–7.4. Pentobarbital (Sigma Aldrich, St. Louis, MO, USA) was dissolved in saline, propylene glycol, and ethanol to make Nembutal. Nembutal was further diluted to make equithesin. Carprofen (Zoetis, Parsipanny, NJ, USA) was diluted in saline. Nicotine, equithesin, and carprofen were filtered via 0.22 μm sterile filters (VWR, Radnor, PA, USA).

### 5.9. Surgical Procedure

On PN 24 or 27, rats were anesthetized with equithesin (0.35 mL/100 g, IP), and catheters were surgically implanted into the right external jugular vein as described previously [62]. After surgery, rats were given the analgesic carprofen (5 mg/kg, SC). Cannulas were flushed daily with sterile heparinized saline solution to maintain patency. All animals were given 3 days to recover from catheter surgery. To determine catheter patency, the day before nicotine-induced locomotor test, rats were administered propofol (Zoetis), a rapid anesthetic (0.05 mL for adolescents, IV).

### 5.10. Acute Nicotine-Induced Locomotor Activity & Open Field Test

On PN 31, animals were given two intravenous injections of nicotine (2 × 0.03 mg/kg/0.1 mL) or saline, spaced 1 min apart. Immediately following the second injection, rats were placed in a novel locomotor chamber to have their activity recorded in 5 min intervals during the total 30 min test.

### 5.11. Sub-Chronic Nicotine-Induced Locomotor Activity & Open Field Test

Adolescent animals were pretreated for 4 days with nicotine (2 × 0.03 mg/kg/0.1 mL) or saline (PN 28–31) [37]. On the fourth day (PN 31), following the last injection of nicotine or saline, rats were placed in a novel locomotor chamber. The locomotor activity was recorded at 5-min intervals during the subsequent 30-min test period.

### 5.12. Tissue Collection & Preparation

At postnatal day PN 31, immediately following nicotine-induced locomotor activity tests, male and female rats were decapitated; then, brains were immediately removed, rapidly frozen in −20 °C isopentane, and stored at −80 °C until use.

#### Tissue Preparation

Twenty-micron sections were cryostat-cut, mounted onto slides at −20 °C, and fixed with 4% paraformaldehyde in 0.1 M PBS, pH 7.4 for 1 h at 22 °C. Slide-mounted tissue were washed in PBS, desiccated, and stored at −20 °C until use.

### 5.13. In Situ Hybridization

cRNA Probe Preparation: [^35^S]-labeled UTP (PerkinElmer) was used to synthesize cRNA riboprobes in the sense and antisense orientation from a pBS SK(−) plasmid containing a 1760 bp fragment of α6 cDNA [63]. cDNA was kindly provided by Dr. J. Boulter (UCLA, Los Angeles, CA).

Hybridization: As previously described by Winzer–Serhan et al. [64], tissue sections were pretreated with Proteinase K (1 µg/mL) for 10 min at 22 °C, acetylated, dehydrated through graded ethanol, and then air dried. Sections were then incubated for 16 h at 60 °C, with hybridization solution (50% formamide, 10% dextran sulfate, 0.02% Ficoll, 0.02% polyvinyl pyrolidone, 0.02% bovine serum albumin, 500 µg/mL tRNA, 10 mM DTT, 0.3 M NAcCl, 10 mM Tris pH 8.0, 1 mM EDTA pH 8.0) containing the [35S]-labeled riboprobe (107 cpm/mL). After hybridization, sections were incubated with RNase A (20 µg/mL) for 30 min at 37 °C and washed twice at 22 °C for 5 min each with 2 × SSC buffer/10 mM DTT and 1 × SSC buffer/10 mM DTT buffer, followed by a 30 min wash in 1 × SSC at 60 °C. Tissue sections were dehydrated and apposed in light-tight cassettes to βmax film with 14C standards of known radioactivity for 12–24 h. Film was developed and rapidly fixed. Sections were postfixed with 4% paraformaldehyde and then processed for Nissl staining using cresyl violet for anatomical analysis.

### 5.14. Statistical Analysis

Data were analyzed with JMP (SAS Institute, Cary, NC, USA). Predefined exclusion criteria for baseline behaviors, nicotine-induced behaviors, and mRNA expression were applied after assessing outliers of box and whisker plot separated by all groups. Animals that did not display immediate anesthesia from propofol were excluded from the analysis.

#### 5.14.1. Baseline Behavior

For food self-administration, mean response data over time were analyzed by a repeated measure four-way analysis of variance (ANOVA) for sex (male/female) × genotype (α6^CC^/α6^GG^) × reinforced/nonreinforced response × day, with a repeated measure on reinforced/nonreinforced response and day. For locomotor activity, mean total ambulatory counts were analyzed by a two-way ANOVA for sex × genotype. For the open field test, mean percent (%) center time data were analyzed by a two-way ANOVA for sex × genotype. For EPM, % open arm time data were analyzed by a two-way ANOVA for sex × genotype. For LDT, % light side time data were analyzed by a two-way ANOVA for sex × genotype. Data for one animal was not recorded for locomotor activity and the open field test; therefore, they were not included in locomotor and anxiety-like behavior analysis. No other animals were excluded for food reinforcement, EPM, or LDT analysis. Bonferroni-corrected post hoc analysis was applied for significant main or interactive effects with one- or two-tailed *t*-tests, as appropriate.

#### 5.14.2. Nicotine-Induced Behaviors

For acute and sub-chronic nicotine-induced locomotor activity, the mean total ambulatory were analyzed by a three-way ANOVA for sex genotype × pretreatment (Saline/Nicotine). One animal was excluded from sub-chronic nicotine analysis based on criteria described above. Acute and sub-chronic nicotine-induced anxiety-like behavior data were analyzed by a three-way ANOVA for sex × genotype × pretreatment for % center time. One animal was excluded from acute and sub-chronic anxiety-like behavior analyses based on criteria described above. Bonferroni-corrected post hoc analysis was applied for significant main or interactive effects with one- or two-tailed *t*-tests, as appropriate.

#### 5.14.3. Autoradiography

Hybridization signal for α6 nAChR subunit riboprobe autoradiographic films were quantified using a video-based computerized image analysis system (MCID, Image Research Inc., St. Catharines, Ontario, Canada). For ISH, the total optical density of the hybridization signal was measured in the substantia nigra (SN), the anterior ventral tegmental area (aVTA), the posterior VTA, and the interpeduncular nucleus (IPN) as identified landmarks, such as emersion of the IPN and the medial terminal nucleus of accessory optical tract (which visibly separated SNc from the VTA in some sections) [65], to delimit the pVTA. The radioactivity values were determined by a standard curve generated from known ^14^C standards and expressed as dpm/mg of tissue. A non-specific signal was determined from the sense probe. Dpm/mg of tissue was analyzed by a three-way ANOVA for sex × pretreatment × genotype. No animals were excluded from analysis.

## Figures and Tables

**Figure 1 ijms-23-03145-f001:**
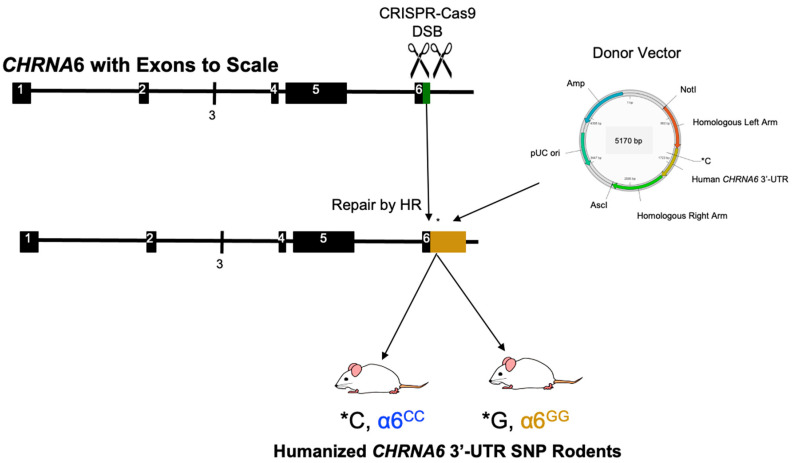
Generation of Humanized *CHRNA6* 3′-Untranslated Region (UTR) SNP Rodents. Site-specific double-stranded breaks (DSB) were introduced at the *CHRNA6* 3′-UTR of the wild-type (WT) Sprague-Dawley (green, 95 nucleotides) via Clustered Regularly Interspaced Short Palindromic Repeats (CRISPR)/CRISPR-associated (Cas9) endonuclease methods followed by the insertion of a donor vector through homologous recombination (HR) DNA repair. A human *CHRNA6* 3′-UTR (gold, 559 nucleotides) containing either the minor or major SNP allele at nucleotide position 123 (SNP location indicated by *C or *G, respectively) was introduced by a donor vector. F2 offspring resulting from confirmed founders and F1 crossbred pups resulted in homozygous major (α6^GG^) and homozygous minor (α6^CC^) humanized *CHRNA6* 3′-UTR SNP rodents. Ampicillin (Amp); base pair (bp); AscI and NotI: restriction enzyme recognition sites; plasmid origin of replication (pUC ori).

**Figure 2 ijms-23-03145-f002:**
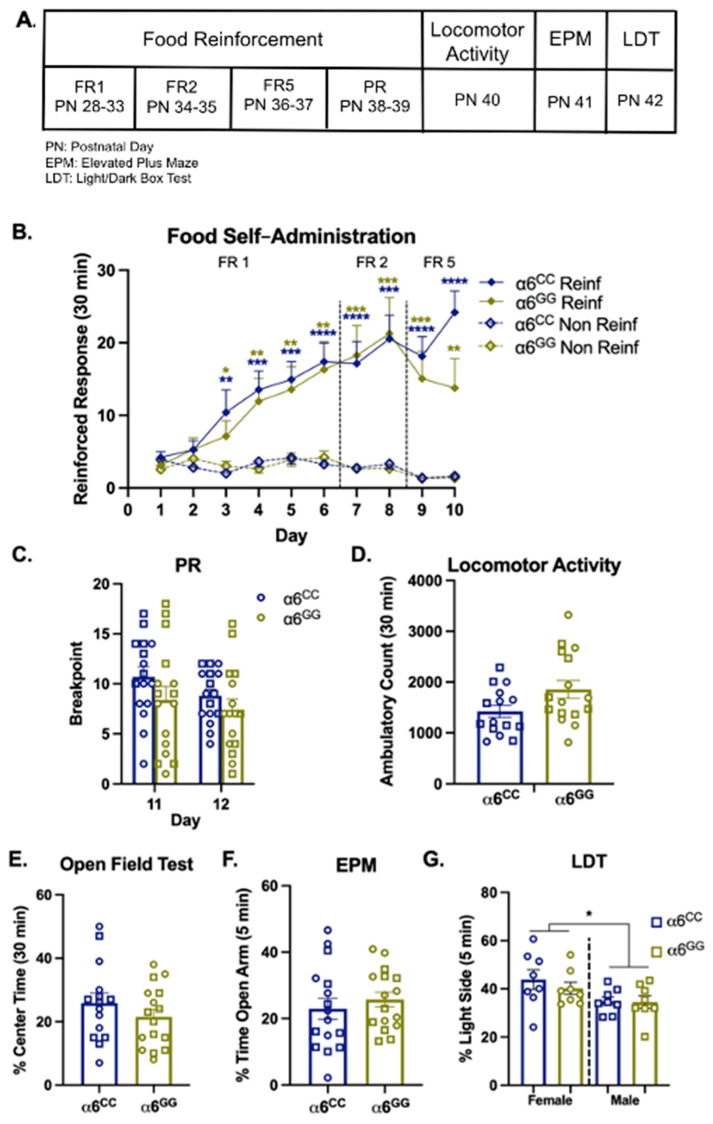
The human *CHRNA6* 3′-UTR knock-in does not impact baseline behavior. (**A**) Experimental paradigm for baseline behaviors. Postnatal Day (PN), Elevated Plus Maze (EPM), and Light/Dark Box Test (LDT). (**B**) Mean daily 30 min response ± SEM for food self-administration at Fixed Ratio (FR)1, FR2, and FR5 schedules of reinforcement, *n* = 16/group. **** *p* < 0.0001; *** *p* < 0.001; ** *p* < 0.01; * *p* < 0.05 Reinforced (Reinf) vs. Non-Reinforced (Non Reinf) responses. (**C**) Mean breakpoint ± SEM for food self-administration under a progressive ratio (PR) schedule of reinforcement, *n* = 16/group. (**D**) Mean total ambulatory counts ± SEM, *n* = 15–16/group. (**E**) Mean percentage (%) of time spent in the center of an open field ± SEM, *n* = 15–16/group. (**F**) Mean % of time spent in the open arm of an EPM ± SEM, *n* = 16/group. (**G**) Mean % of time spent on the light side of a Light/Dark box ± SEM, *n* = 8/group. * *p* < 0.05 females vs. males. Circles represent females and squares represent males.

**Figure 3 ijms-23-03145-f003:**
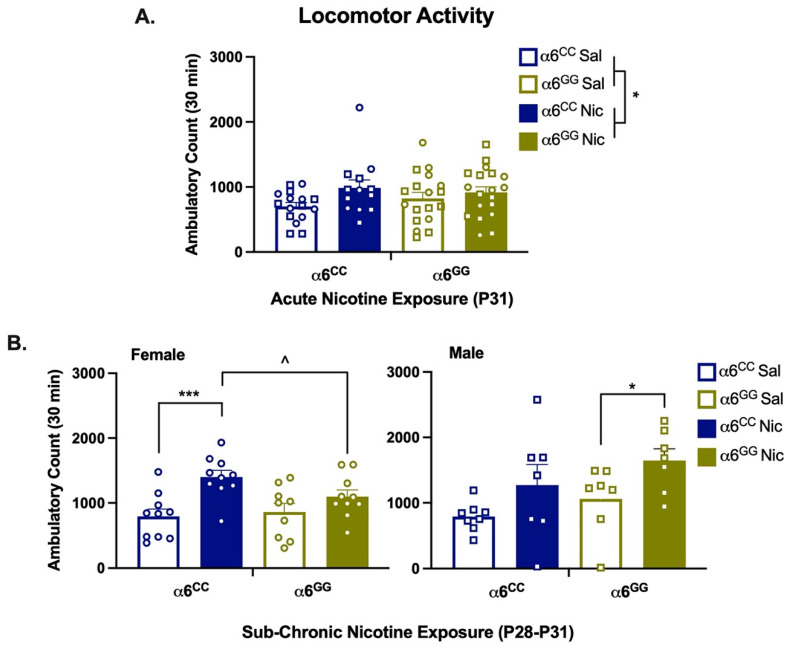
Nicotine-induced locomotion in humanized *CHRNA6* 3′-UTR SNP rats. Mean total ambulatory counts ± SEM for (**A**) 1-day nicotine exposure, collapsed by sex *n* = 13–19/group. (**B**) 4-day nicotine exposure, * *p* < 0.05, *** *p* < 0.001 Nic vs Sal; ^ *p* < 0.05 α6^CC^ Nic vs α6^GG^ Nic *n* = 7–10/group. Saline (Sal), Nicotine (Nic). Circles represent females, and squares represent males.

**Figure 4 ijms-23-03145-f004:**
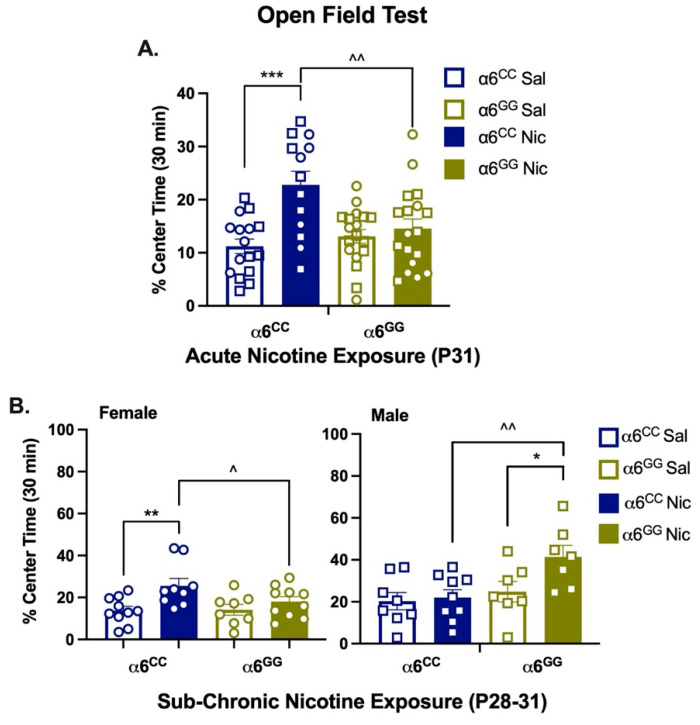
Nicotine-induced anxiolytic behavior in humanized *CHRNA6* 3′-UTR SNP rats. Mean percent (%) time spent in center ± SEM for (**A**) acute (1-day) nicotine exposure *** *p* < 0.001 α6^CC^ Nic vs α6^CC^ Sal, ^^ *p* = 0.01 α6^CC^ Nic vs α6^GG^ Nic *n* = 13–18/group. (**B**) Sub-chronic (4-day) nicotine exposure, * *p* < 0.05, ** *p* < 0.01 Nic vs Sal; ^ *p* < 0.05, ^^ *p* < 0.01 α6^CC^ Nic vs α6^GG^ Nic, *n* = 7–10/group. Saline (Sal), Nicotine (Nic). Circles represent females, and squares represent males.

**Figure 5 ijms-23-03145-f005:**
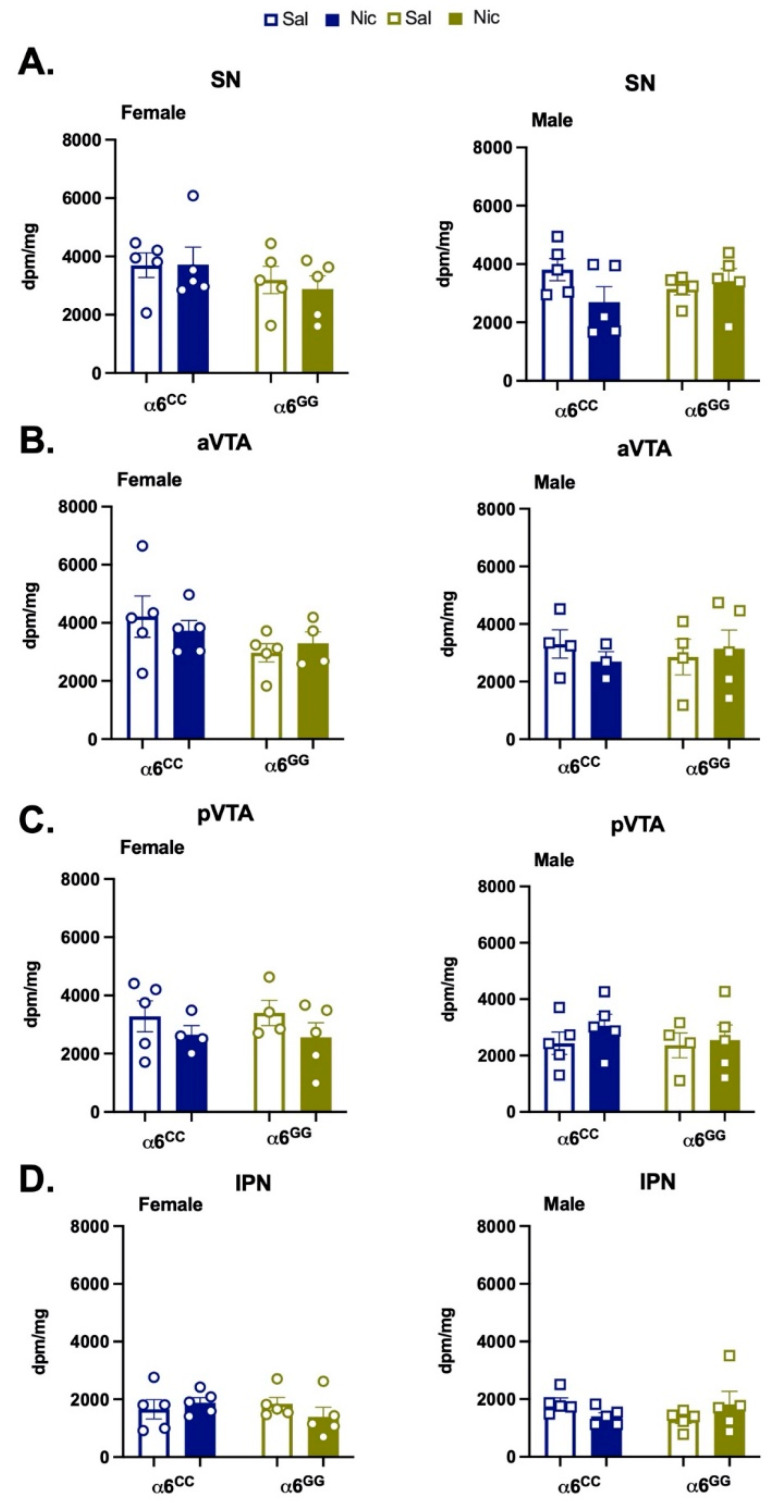
Nicotine does not alter α6 nicotinic receptor subunit mRNA expression in humanized *CHRNA6* 3′-UTR SNP Rats. Mean dpm/mg ± SEM for sub-chronic nicotine exposed male and female rats in the (**A**) substantia nigra (SN), (**B**) anterior ventral tegmental area (aVTA), (**C**) posterior ventral tegmental area (pVTA), and (**D**) interpeduncular nucleus (IPN), *n* = 3–5/group. Saline (Sal), Nicotine (Nic). Circles represent females, and squares represent males.

## Data Availability

The data presented in this study are available on request from the corresponding author.

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
