# Peer review of "Sex- and Genotype-Dependent Nicotine-Induced Behaviors in Adolescent Rats with a Human Polymorphism (rs2304297) in the 3′-UTR of the CHRNA6 Gene"

_ijms, 2022, doi:10.3390/ijms23063145_

Round 1

Reviewer 1 Report

The original article written by Cardenas A et al., focuses on studying the effects of the human CHRNA6 3’-UTR SNP, through the generation of knock-in rodent lines with either C or G SNP alleles, regarding tobacco and substance use in the adolescent human population.  The Authors investigate the hypothesis that the human CHRNA6 3’-16 UTR SNP knock-in does not impact baseline but enhances nicotine-induced behaviors through the performance of rodent behavioural tasks. The manuscript is well written and the pictures are suitable to follow the different results the authors provided. The discussion and conclusion suggest that the human CHRNA6 3’-UTR SNP of their lab is functional in their in vivo model, bringing interesting new insights about this matter. Thus, the article is suitable to be published but after some minor revisions, listed here below:

- Please check double blank spaces (i.e. line 107). 

- p of the p-value is or is not written in italic? Please choose one form and correct accordingly. 

- The corresponding author has to be indicated with * also nearby the related author name.

- Abstract: Please specify what e-cigarette and CHRNA6 mean.

- Has CHRNA6 as keyword to be in italic font?

- Line 30: please specify what e-cigarette means.

- Line 47: “substance tried” is not so clear. Please formulate the sentence again.

- Line 60: Is α6β2* correct or should it be α6* β2?

- Line 79: Please revise the word Creation.

- Figure 1: Please indicate in the figure legend what the acronyms mean.

- Line 109: an s in show is missing.

- Figure 2: please check the resolution of this figure and modify accordingly.

- Please check the style of the figure legend and correct accordingly.

- Paragraph 2.5: please describe the tissues involved in the analysis and why did you investigate them also in the description of these results, not only cited them in the related figure legend.

Author Response

- Please check double blank spaces (i.e. line 107).

  • We have removed double blank spaces

- p of the p-value is or is not written in italic? Please choose one form and correct accordingly.

  • Thank you, p of p-value is now uniform throughout the manuscript

- The corresponding author has to be indicated with * also nearby the related author name.

  • We have indicated the corresponding author with an *

- Abstract: Please specify what e-cigarette and CHRNA6 mean.

  • The abstract has been edited. E-cigarette has been removed. CHRNA6 has been specified as a nicotinic receptor subunit gene.

- Has CHRNA6 as keyword to be in italic font?

  • Italicized CHRNA6 in keywords

- Line 30: please specify what e-cigarette means.

  • We have specified what e-cigarette means.

- Line 47: “substance tried” is not so clear. Please formulate the sentence again.

  • The sentence has been changed to: “The α6 nAChR subunit SNP is associated with increased cigarette smoking and drug experimentation during adolescence [12-17]”.

- Line 60: Is α6β2* correct or should it be α6* β2?

  • For clarity, we changed α6β2* to α6β2-containing.

- Line 79: Please revise the word Creation.

  • Creation was changed to Generation

- Figure 1: Please indicate in the figure legend what the acronyms mean.

  • All acronyms have been defined.

- Line 109: an s in show is missing.

  • An s has been added to the word show on what is now line 145.

- Figure 2: please check the resolution of this figure and modify accordingly.

  • We have included the 300dpi resolution figure in the manuscript

- Please check the style of the figure legend and correct accordingly.

  • Uncertain what the reviewer means by style of figure legend.

- Paragraph 2.5: please describe the tissues involved in the analysis and why did you investigate them also in the description of these results, not only cited them in the related figure legend.

  • In addition to describing the regions in the figure legend and statistical analysis of materials and methods, we have included the rationale for assessing the regions of interest and list them in section 2.5.

Reviewer 2 Report

The proposed topic is very interesting and could provide interesting present and future ideas. However there are some major revisions needed.
Abstract.
The abstract must be reorganized according to this scheme: 2 lines of introduction, study objective, methods, results, conclusions, respecting the appropriate number of words.

Introduction.
The introduction is well argued and clear. The Authors in the final part of the introduction should insert the objective of the study and the reason of interest of the manuscript for the scientific community.

Methods.
Why do the authors insert methods after the conclusions? Authors should move methods after introduction. The methods are described very well.

Results.
Properly described and clear.

Discussion
"Baseline behaviors" paragraph.
Why are the findings of this study contrasting with previous ones? Authors should provide an explanation.

Conslusion
Clear and concise, however the authors should cite some useful studies to define the concept of addiction genotyping.
1) Di Nunno, N., Esposito, M., Argo, A., Salerno, M. and Sessa, F., 2021. Pharmacogenetics and Forensic Toxicology: A New Step towards a Multidisciplinary Approach. Toxics, 9 (11), p.292.
2) Orban, G .; Bombardi, C .; Marino Gammazza, A .; Colangeli, R .; Pierucci, M .; Pomara, C .; Pessia, M .; Bucchieri, F .; Benigno, A .;
Smolders, I .; et al. Role (s) of the 5-HT2C receptor in the development of maximal dentate activation in the hippocampus of
anesthetized rats. CNS Neurosci. Ther. 2014, 20, 651–661.
3) Pirmohamed, M. Pharmacogenetics: Past, present and future. Drug Discov. Today 2011, 16, 852–861
4) Gilbert, D.G., Izetelny, A., Radtke, R., Hammersley, J., Rabinovich, N.E., Jameson, T.R. and Huggenvik, J.I., 2005. Dopamine receptor (DRD2) genotype-dependent effects of nicotine on attention and distraction during rapid visual information processing. Nicotine & Tobacco Research, 7 (3), pp. 361-379.
4) Gieryk, A., Ziolkowska, B., Solecki, W., Kubik, J. and Przewlocki, R., 2010. Forebrain PENK and PDYN gene expression levels in three inbred strains of mice and their relationship to genotype-dependent morphine reward sensitivity. Psychopharmacology, 208 (2), pp. 291-300.

Author Response

Abstract.

The abstract must be reorganized according to this scheme: 2 lines of introduction, study objective, methods, results, conclusions, respecting the appropriate number of words.

  • The abstract has been edited to include the study objective and follow the organization suggested within the word limit of the journal.

Introduction.

The introduction is well argued and clear. The Authors in the final part of the introduction should insert the objective of the study and the reason of interest of the manuscript for the scientific community.

  • The objective of the study and reason for interest for the scientific community has been added to the introduction

Methods.

Why do the authors insert methods after the conclusions? Authors should move methods after introduction. The methods are described very well.

  • IJMS author guidelines for manuscript preparation place materials and methods after the discussion

Discussion - "Baseline behaviors" paragraph.

Why are the findings of this study contrasting with previous ones? Authors should provide an explanation.

  • We have included an explanation for why our observations contrast with that of the other study.

Conclusion

Clear and concise, however the authors should cite some useful studies to define the concept of addiction genotyping.

  • We thank the reviewer for these insightful studies to frame our findings in the scope of pharmacogenetics. We have integrated pharmacogenetics and such studies into the manuscript.

Reviewer 3 Report

In a manuscript entitled "Sex- and genotype-dependent nicotine-induced behaviors in adolescent rats with a human polymorphism (rs2304297) in the 3’-UTR of the CHRNA6 gene" authors present and validate a novel, humanized rodent model to study the effects a human SNP in the 3’-UTR of the CHRNA6 gene in nicotine-induced locomotor activity and anxiety-like behavior. Manuscript describes at the appropriate manner the purpose of the work, techniques and methods used, major findings with important data and conclusions. Obtained results highlight the nuances and importance of including age, biological sex, environmental influences, and genetics to evaluate nicotine use.

I have no additional corrections or suggestions on the manuscript.

Author Response

I have no additional corrections or suggestions on the manuscript.

  • We thank the reviewer for their time and review.

Round 2

Reviewer 2 Report

The paper has been improved enough for the journal